# A Critical Review of the Time-Dependent Performance of Polymeric Pipeline Coatings: Focus on Hydration of Epoxy-Based Coatings

**DOI:** 10.3390/polym13091517

**Published:** 2021-05-09

**Authors:** Hossein Zargarnezhad, Edouard Asselin, Dennis Wong, C. N. Catherine Lam

**Affiliations:** 1Department of Materials Engineering, The University of British Columbia, 309-6350 Stores Road, Vancouver, BC V6T 1Z4, Canada; edouard.asselin@ubc.ca; 2Shawcor Ltd., 25 Bethridge Road, Toronto, ON M9W 1M7, Canada; dennis.wong@shawcor.com (D.W.); catherine.lam@shawcor.com (C.N.C.L.)

**Keywords:** powder-based coatings, glassy epoxy, mass transfer properties, wet-state diffusion, permeability, coating degradation

## Abstract

The barrier performance of organic coatings is a direct function of mass transport and long-term stability of the polymeric structure. A predictive assessment of the protective coating cannot be conducted a priori of degradation effects on transport. Epoxy-based powder coatings are an attractive class of coatings for pipelines and other structures because application processing times are low and residual stresses between polymer layers are reduced. However, water ingress into the polymeric network of these coatings is of particular interest due to associated competitive sorption and plasticization effects. This review examines common analytical techniques for identifying parameters involved in transport in wet environments and underscores the gaps in the literature for the evaluation of the long-term performance of such coating systems. Studies have shown that the extent of polymer hydration has a major impact on gas and ion permeability/selectivity. Thus, transport analyses based only on micropore filling (i.e., adsorption) by water molecules are inadequate. Combinatorial entropy of the glassy epoxy and water vapor mixture not only affects the mechanism of membrane plasticization, but also changes the sorption kinetics of gas permeation and causes a partial gas immobility in the system. However, diffusivity, defined as the product of a kinetic mobility parameter and a concentration-dependent thermodynamic parameter, can eventually become favorable for gas transport at elevated temperatures, meaning that increasing gas pressure can decrease selectivity of the membrane for gas permeation. On the other hand, reverse osmosis membranes have shown that salt permeation is sensitive to, among other variables, water content in the polymer and a fundamental attribute in ionic diffusion is the effective size of hydrated ions. In addition, external electron sources—e.g., cathodic protection potentials for pipeline structures—can alter the kinetics of this transport as the tendency of ions to dissociate increases due to electrostatic forces. Focusing primarily on epoxy-based powder coatings, this review demonstrates that service parameters such as humidity, temperature, and concentration of aggressive species can dynamically develop different transport mechanisms, each at the expense of others. Although multilayered coating systems decrease moisture ingress and the consequences of environmental exposure, this survey shows that demands for extreme operating conditions can pose new challenges for coating materials and sparse data on transport properties would limit analysis of the remaining life of the system. This knowledge gap impedes the prediction of the likelihood of coating and, consequently, infrastructure failures.

## 1. Introduction

Many coating technologies have been developed to fulfill industrial demands for protective barriers for steel structures [1]. However, coating technologies face increasing challenges posed by high or low operating temperatures as well as new requirements for increased longevity and abrasion resistance [2]. Epoxy-based coatings are commonly used for oil and gas pipelines and other steel structures [3]. In addition to good performance for corrosion protection and resistance to weather, and humidity [4], epoxy-based powder coatings have a significantly shorter curing time relative to liquid and thermoplastic coatings—they reduce processing time while providing optimal anticorrosion performance [5]. The few cases of early powder coating failure documented in the literature are mostly due to inappropriate substrate pretreatment, low application temperature, improper curing, or improper chemistry in the formulation [6].

The development of an organic coating requires compatibility with specific environmental regulations and safety concerns [7]; the coating’s molecular structure is modified in some cases to meet new expectations [8,9,10]. For instance, the peripheral polar functional groups in the epoxy act as adsorption sites for inhibitors (e.g., amines, thiols, and alcohol-based epoxy resins), which enhances its anticorrosive effects in aqueous media when applied as a coating [11]. In addition to the main macromolecular compound, organic and inorganic additive components can be used to improve the anticorrosive performance of the coating [12]. In general, functional components in a coating formulation serve different purposes: solvent to control film adhesion/formation; organic or polymeric binders to provide barrier properties; dispersed pigments/functional fillers to enhance mass transport properties and UV resistance; fillers to provide uniformity, and other additives to inhibit substrate corrosion [4,13]. There are many proposed new coating systems for petroleum-based products and, since climate change is a serious concern, recent trends in alternative fuels suggest that there is a necessity to increase the bio-based material content in coatings [14,15,16]. However, the pipeline industry continues to use epoxy-based coatings. Some functional fillers such as needle-shaped amorphous wollastonite are used to increase flexural modulus and lower the thermal expansion and shrinkage of the final coating [17,18]. Microencapsulated agents―also known as self-healing components―may also be used as additives. These provide a polymer mending property in case of mechanical damage to the coating and may effectively protect the coated steel surface from corrosion [19,20].

Environmental parameters such as humidity, temperature, operating pressure, and ageing can individually or in combination induce various complex mass transport scenarios [21,22]. The interactions between the varied components of the coating and the eventual service conditions are central to the long-term performance of the coating. The mass transport properties of the coating provide a metric for this interplay. Effects of the coating’s structural attributes—e.g., its polarity, chain stiffness, and inertness to the penetrant—on the extent of permeation are discussed in the literature [23], however, data on transport rates of various species and their competitive permeation through coatings are scant.

Fusion-bonded epoxy (FBE) is a thermoset polymer with excellent long-term adhesion performance at moderate temperatures (e.g., 65 °C) [19] and is often preferred for both internal linings and external coatings on pipelines [24,25]. Outside a few reports on individual field-applied coatings, there are no standard data for the mass transfer properties of FBE [26,27], possibly because of the specificity of the diverse FBE formulations to various applications [28]. In severe environments, additional polyolefin layers may be applied over an FBE primer to reinforce the protection against corrosive species [29,30,31]. High-Performance Powder Coating (HPPC) is a monolithic coating structure with efficient interlayer compatibility, and is qualified for service under aggressive environments [32,33]. Although excellent results have been reported, demands for aggressive working conditions (e.g., wet-state applications, high temperatures, oxygen, and salts) may pose new challenges for these protection systems [34]. By focusing on these technologies as exemplars of single- and multi-layered coatings in the present study, we outline challenges involved in gaining a predictive approach for analysis of a given coating’s long-term performance.

Water plays a significant role in the dynamics of epoxy degradation and affects the mass transfer rate of different permeants in the coating [35,36]. Mass transport analysis to quantify permeation data through FBE and HPPC in wet-state applications is inadequate. An objective of this survey is to outline the existing literature in mass transport characterization of coating systems. Thus, analytical studies on epoxy resins and analogous glassy polymers are reviewed in the following subsections. The interconnections between mass transport and the coating degradation are also identified. Further, reports on the impacts of protective measures like cathodic protection (CP), which may affect mass transport and barrier protection by the coating, are briefly reviewed. We then present an overview of transport properties for related polymeric structures that are associated with powder coating systems and explore empirical methods to assess the permeability through multilayered coatings. Lastly, we briefly discuss requirements to achieve a systematic modelling scheme for the time-dependent performance of an a priori defect-free coating system that is based on mass transport, declining properties, and service parameters.

## 2. Mass Transport through Coating Materials

Mass transfer characteristics of coatings have received less attention than post-disbondment events, especially for pipeline materials in which long-term integrity is of major interest. Failure analysis of pipeline corrosion has shown that the most common forms of coating defect are holidays and cathodic disbondment (CD) [37,38], and thus, significant progress has been made to enhance the durability of organic coatings against these issues. However, environmental effects on operating pipeline coatings, such as moisture and temperature, are understudied or unexplored. Among components present in the coating formulation, binders are expected to facilitate water and oxygen transmission [39] and, consequently, could contribute to matrix deterioration. Permeabilities of a coating system against water, oxygen, and ionic species are fundamental attributes necessary to identify rate-determining steps of corrosion of the underlying steel substrate [23]. According to the solution-diffusion model, transmission of permeant molecules through polymeric coating films (expressed as a permeability coefficient) occurs in a three-step process [40]:Dissolution in the polymer (determined by the solubility coefficient) from the exposed sideDiffusion from higher to lower concentration/pressure (determined by diffusion coefficient)Desorption from the other side of the polymer film

Permeability is thus a function of solubility in the coating and diffusivity through the coating, either of which may dominate depending upon interactions between the polymer and the permeant. Generally, diffusivity dominates in gaseous permeation, unless the gas is easily condensable (e.g., CO_2_) or is water vapor [41]. When a coating system is required to prevent simultaneous ingress of gases and liquid water, data from individual species permeability tests (e.g., oxygen permeability in the absence of water vapor) may not adequately represent barrier properties. In the following sections, a background on water transport through epoxy is provided and empirical approaches to analyze impacts of hydration on associated coating structures are outlined. Then, gaps in the existing literature data on gaseous and ionic transport in the wet-state condition will be reviewed.

### 2.1. Water Ingress into an Epoxy Network

Water molecules are highly prone to interacting with existing polar sites, typically on the order of 2–20 Å in diameter, in a continuous network of nanopores in the epoxy structure [42,43]. Hence, the nanopores increase both the active polarity of, and moisture uptake by, the network. Subsequently, if molecular degrees of freedom are high enough, free water molecules can also open up the structure, reorganizing the molecular topology (i.e., plasticization) and revealing more polar sites in which moisture can reside [42,44]. Zhou and Locus classified the behavior of water molecules binding with epoxy into two types:when they form a single, easy-to-remove hydrogen bond with hydrophilic groups in the epoxy network; andwhen they form multiple, stronger bonds in the epoxy network [45].

Type I behavior breaks the interchain Van der Waals force in epoxy and decreases its glass transition temperature (*T_g_*), while Type II forms a secondary crosslink network and lessens *T_g_* depression [46]. Cured epoxy resins with different side groups (i.e., different fractional free volume) were shown to differ in terms of changes of free volume size and magnitude of increase in diffusion coefficient when temperature was raised from 20 to 80 °C [47]. Type II water molecules were shown to form bonds (release heat) before Type I molecules and Zhou and Locus proposed a mechanism to explain resin plasticization in epoxy: local-chain mobility of the polymer network hinders the dissociation of Type II bound water molecules, whereas lower desorption activation energy for Type I facilitates forming a single hydrogen bond with epoxy resin. Therefore, Type I bound water disrupts the interchain Van der Waals force and diffuses into the epoxy network resulting in swelling and plasticization [45].

From a materials design perspective, the network architecture can affect the diffusion and absorption properties of epoxy [48,49,50]. Flexibility (or segmental conformation) of epoxy side groups is unfavorable for water absorption, but favorable for its transport: the exothermic interaction between water and polar sites is smaller for more flexible functional groups [51]. Crosslinking generally reduces the free volume, swelling, diffusivity and water absorption in coating films [35]. However, the tendency of the coating to retain the already absorbed water may increase at higher crosslinking densities; this will result in defects, such as phase separation, blister formation, and cracking [52]. Nanoscale mapping of water absorption also confirms that bulk equilibrium uptake is localized to the highly cross-linked regions in epoxy coatings [53].

Sorption and transport of a penetrant into a polymer network can isothermally change the state of the solid phase from a glassy, highly viscous brittle structure, to a rubbery, less viscous and more mobile structure [54]. According to the Flory-Huggins theory [55,56,57], the thermodynamic state of a penetrant/polymer system is expressed by the combinatorial entropy of the mixture; glassy and rubbery contributions are then related to the stiffness of the dry polymer sample (characterized e.g., by *T_g_*) and the quantity of sorbed penetrant. At steady-state conditions, using Fick’s first law of diffusion, one can relate the permeability of penetrant *i*, *P_i_* (mol/m-s-Pa), to diffusion coefficient, *D_i_* (m^2^/s), and solubility in the polymer, *S_i_* (mol/m^3^-Pa) [58]:(1)$$P_{i}=D_{i}\times S_{i}$$

Because solubility has a thermodynamic origin and depends on polymer-permeant interactions as well as on the gas condensability, Equation (1) shows that an entropy change in the polymeric solvent due to the mass transfer (determined by $S_{i}$) can dynamically affect the permeant transmission. The concentration of the permeant in the polymer is defined by Henry’s Law as the product of the solubility and the partial pressure of gas, assuming low gas partial pressures (<1 atm). At higher partial pressures, thermodynamic consideration of whether the polymer is rubbery or glassy, combined with plasticization and swelling effects, results in concentration-dependent permeability estimates [59].

#### 2.1.1. Water Transport: Gravimetric Analysis

Classic gravimetric methods make it viable to focus on each of the parameters expressed in Equation (1) [44,47]. Water uptake ($w_{u})$, defined as the mass fraction of water in the hydrated polymer, is usually calculated from the mass gained by the polymer in an immersion test over the dry polymer mass. It is then used to calculate the equilibrium volume fraction of water, $v_{W}$, as follows [60]:(2)$$v_{W}=\frac{w_{u}}{w_{u}+\rho_{w}/\rho_{p}}$$
where $\rho_{w}$ and $\rho_{p}$ are densities (g/cm^3^) of water and dry polymer, respectively. Data from water uptake measurements as a function of time also correlate to the diffusion coefficient based on the solution for Fick’s second law. At very short times, when the time-dependent moisture uptakes ($w_{t}$) are less than half of the equilibrium moisture uptake ($w_{\infty }$), the Fickian diffusion rate is much less than that of polymer segment mobility, and *D_i_* is approximated by the following:(3)$$D_{i}=\frac{\pi }{16}{\left(\frac{w_{t}/w_{\infty }}{\sqrt{t}/l}\right)}^{2}$$
where *l* and *t* are the sample thickness (m) and time (s), respectively. Different polymeric structures and compositions feature myriad arrangements of functional groups and unique free volume distributions, leading to variable water diffusion behaviors (Table 1). This ideal Fick’s process assumption (i.e., Equation (3)) ignores some crucial factors such as polymer changes as a result of water diffusion and the true effect of temperature on the activation energy of diffusion [61]. Apart from the addressed limitations, Table 1 suggests that the order of magnitude for the self-diffusion coefficient of water vapor in FBE is 10^−13^–5 × 10^−12^ m^2^/s.

The water vapor transmission (WVT) test is an industry-approved cup method to quantify the permeation rate through a coating film [64,65]. Despite errors associated with elevated temperatures because of film deformation and damage caused by pressure fluctuations [66], a quantitative assessment of equilibrium water transmission rates can result from these measurements (Table 2). Limitations to the method are potential sources of error in cup weights [67] and the need for freestanding films with uniform thickness [68]. The latter may not be feasible when assessing the permeability of powder coatings. Results for permeant *i* are most commonly reported in permeance ($P_{i}/l)$, similar to what is done for gas permeation cells [69]:(4)$$\left(\frac{P_{i}}{l}\right)\left(\mathrm{mol}/m^{2}\cdot \mathrm{Pa}\cdot s\right)=\frac{J}{A\left(p_{fi}-p_{pi}\right)}$$
where l (m) is the coating thickness, *J* (mol/s) is permeant flux, *A* (m^2^) is the geometrical test area, *p_fi_* and *p_pi_* are absolute pressure (Pa) of permeant on the feed and permeate side, respectively. When coatings of different thicknesses are studied at temperatures beyond specified standard ranges (e.g., above 65 °C) [70], reporting mass transport independently of thickness (i.e., permeability or *P_i_*) is most appropriate [71]. Accordingly, reporting WVT results with additional permeability data can be more informative than a qualitative assessment (e.g., [70,72]) centered on permeances through different coating systems which have different thicknesses.

Permeability is sensitive to concentration (i.e., vapor pressure) for hydrophilic polymers like epoxy, and it may profoundly affect the permeability measurement, especially at higher temperatures (Figure 1) [67,74]. Cup methods are used in WVT tests which generally establish 0% or 100% relative humidity (RH) on one face of the coating sample (dry- or wet-cup limits, respectively).

Weight changes of the cups after exposure to a controlled RH at a constant temperature are then measured, which result in average permeability values. It is possible, however, to construct a spot permeability curve based on average permeability values and RH limits that generates equal areas between the curve and the limits—i.e., the dotted curve in Figure 1 has been plotted based on ${\bar{P}}_{1}$ and ${\bar{P}}_{2}$ from dry and wet cup tests, respectively ($area_{A}=area_{B}\;\mathrm{and}\;area_{C}=area_{D}$) [67,75]. The resulting spot permeability curve can then be a dependable reference to determine average permeability between two individual humidity limits across the coating film. For example, to determine $\bar{P}$ across an applied coating film at 65% RH from the spot permeability curve in Figure 1, one can find a horizontal line between 0 and 65% RH where equal areas between the curve and these limits are generated.

Water exists in three states in hydrophilic polymers: (i) non-freezable bound water, (ii) freezable bound water, (iii) free water. The non-freezable type does not crystallize even when the temperature is lowered to −100 °C, while freezable bound and free water crystallizes under and at 0 °C, respectively [50]. These different states reflect the intermolecular interactions between polymer polar sites and water molecules absorbed through a transport process. Popineau et al. [76] combined gravimetric analysis and nuclear magnetic resonance (NMR) signals to detect two types of water molecules (i.e., ‘mobile’ and ‘bound’) and proposed a four-step mechanism for water diffusion in epoxy resin above its *T_g_*. Conventional gravimetric water analysis studies can effectively explain free water permeation (i.e., WVT tests) and partially explain total water sorption (i.e., immersion tests). However, they are limited in the information they provide and cannot explain the deterioration of barrier properties as a function of water sorption and time.

Diffusivity of water through a polymeric material can also be measured following an analysis of the desorption process: mass loss data of a hydrated polymer vs. square root of time can indicate *D* (i.e., Equation (3)). A useful analytical method to monitor water desorption as a function of temperature is thermogravimetric analysis (TGA) [77]. This method generates sufficient data for a mechanistic explanation, especially for studying the kinetics of thermal degradation of crosslinked polymers such as epoxy [78,79,80,81]. The weight loss due to physicochemical changes in a material is usually combined with infrared analysis to study water content by spectroscopic means [50,82]. The difference between the net weight loss of epoxy samples soaked at 25 and 70 °C suggests a temperature-dependent absorption of water in the epoxy structure [83]. In addition, hydrated samples in [83] could not be dried completely under high vacuum at their soaking temperature for extended periods of time and no weight loss happened until the temperature surpassed the resin cure temperature. This means the epoxy network is subjected to a chemical transformation upon hydration and the extent of damage may not be eliminated by water removal from the epoxy structure.

#### 2.1.2. Water Transport: Infrared (IR) Spectroscopy Methods

IR techniques offer an advantage over gravimetric techniques in that they provide quantitative information with which to study coating changes. For example, the type of water (free or bound) in the epoxy network can be identified and related to associated physical interactions (e.g., hydrogen bonding) or chemical reactions (e.g., polymer degradation) [84,85]. However, the peaks denoting bound water in epoxy are found in the near IR range, which can be technically difficult to interpret due to the appearance of other coinciding factors [86]. In general, degradation of the epoxy structure by free and bound water is tracked using two mid-range peaks at 3610 and 3370 cm^−1^, respectively. Similar to moisture uptake analysis to measure diffusivity, absorbance data from IR spectroscopy can be used to calculate the water diffusion coefficient from the solution of Fick’s second law (i.e., Equation (3)) [87,88]. Li et al. [89] identified four types of water molecules in IR spectra based on their associated bonding with the epoxy network: both free and bound water generate a vibration at 3472 and 3210 cm^−1^, respectively (Figure 2). Li et al. postulated that water diffusion in epoxy is primarily achieved by water molecules with loose hydrogen bonding to weak polar groups (e.g., sulfone or tertiary nitrogen atoms) or to strong polar groups but under steric hinderance. In the near IR range, the combination of the asymmetric stretching and in-plane deformation mode as a result of water uptake produces a characteristic peak at 5215 cm^−1^; quantitative interpretation of this peak to differentiate the strength of various hydrogen bonds is difficult [90]. Hydroxyl vibrations in the range 7800–6000 cm^−1^ can also be deconvoluted into three peaks at 7075, 6820, and 6535 cm^−1^, which are quantitatively attributed to free water, an intermediate hydrogen-bonded water, and bound water, respectively [82]. It should be noted that these structural changes happen prior to swelling of the polymer: O–H stretching in the 3100–3610 cm^−1^ region decreases upon swelling [91].

Mid-range IR studies on hydrothermally aged FBE (at 85 °C) show that water uptake increases linearly within three months until *T_g_* reaches a depression point and then plateaus [92]. This indicates that, depending on the thermal conditions of the system, the proportion of water in the epoxy that is non-freezable can increase over time. Using in-situ time-resolved IR measurements on an FBE-coated Si substrate, Nguyen and Martin [93] showed that a 50 nm thick layer of water forms after 100 h of exposure to distilled water at 24 °C. Although water did enter and pass through the FBE, forming a layer many monolayers thick on the Si substrate, it remains uncertain that the same interfacial water uptake can develop in an FBE/grit blasted steel system—for example, the findings of Nguyen and Martin as to the interfacial water uptake might be related to moisture adsorption on the Si substrate prior to coating application. A similar technique was also applied to study the effect of water traps (i.e., particles with high water sorption capacity and low water diffusion coefficient) on water diffusion through a sandwich-structured epoxy composite—the data was used to optimize the design for corrosion performance improvement [94]. Monitoring the water band intensity and the interfacial water uptake in this study also showed no time lag for the sorption process (i.e., the elapsed time before water reaches the surface of the reflection element) in both pure and composite epoxy. The initial permeation was attributed to micro-pores formed by solvent volatilization during curing.

#### 2.1.3. Water Transport: Electrochemical Impedance Spectroscopy (EIS)

The foundation for the EIS measurement of water uptake is based on capacitance changes of the coating layer during absorption [95]:(5)$$\varphi \left(\%\right)=\frac{\log\left(C_{t}/C_{0}\right)}{log\left(\varepsilon_{w}\right)}\times 100$$
where φ(%) is the water volume fraction in the coating, $\varepsilon_{w}$ the temperature-dependent relative dielectric constant of water (e.g., 78.3 at 25 °C), and $C_{t}$ and $C_{0}$ are coating capacitance (F/m^2^) at time *t* and zero, respectively. By combining the gravimetric and capacitance methods, it is possible to correlate φ to the amount of water taken up ($M_{t}/M_{\infty }$) and thereby obtain *D* [96]. However, this approximation is only valid for randomly distributed spherical water inclusions; finite element analysis showed that cylindrical inclusions can produce a much greater change in capacitive response relative to spherical ones because the formation of percolation paths is more possible [97]. Therefore, a simple application of Equation (5) may overestimate the true water content regardless of inclusion geometries [98,99]. It should also be noted that Equation (5) assumes constant geometrical parameters for an epoxy coating over time and thereby ignores swelling effects. Introducing the time-dependent thickness to include swelling effects in the analysis yields the same results as measured by gravimetric methods—i.e., modifying Equation (5) with a corrected capacitance $C_{t}.d\left(0\right)/d\left(t\right)$ instead of $C_{t}$ where d(0) and d(t) are thickness at time zero and *t*, respectively [100,101].

When the coating is applied on a metal substrate, the support from the substrate metal can affect barrier properties as well as other parameters such as internal stresses. Accordingly, EIS is a dependable technique since it can provide information on barrier performance of a coating/metal system for exposures to long-term ageing conditions [102]. A comparison of attached free films and applied coatings using EIS analysis showed that a salt solution electrolyte requires remarkably more time to reach the coating/metal interface in the latter case, which implies that adhesion can affect the coating degradation and corrosion rate of the underlying substrate to a great extent (up to one order of magnitude) [103]. Stresses and distortions resulting from water uptake are accompanied by deformation of the free coating film; polymeric relaxation due to water plasticization leads to film deformation, which then develops pores and swollen regions capable of retaining water [104]. Water diffusion analysis based on EIS measurements for an epoxy coating on a galvanized steel substrate shows that the diffusivity of water increases from 0.35 × 10^−13^ m^2^/s at 20 °C to 1.47 × 10^−13^ m^2^/s at 65 °C [105]. These data are relatively lower than those obtained from free films using gravimetric sorption measurements (i.e., 1.04 × 10^−13^ m^2^/s at 22 °C to 8.54 × 10^−13^ m^2^/s at 60 °C) [63]. This comparison confirms two facts regarding quantitative water ingress analysis: (1) gravimetric analysis usually results in an upper boundary for mass transport parameters [35], especially for diffusivity measurements, and (2) coating/substrate adhesion forces control the water uptake by the coating [103]. Once disbondment takes place at the coating/substrate interface, corrosion reactions become involved and water mass transport through the coating shifts toward the upper boundary values more akin to those found through free films.

### 2.2. Vapor/Gas Transport

Conducting gas permeability measurements requires a steady state flow of the permeating gas across the polymeric membrane; the traditional time-lag method remains popular due to its simplicity and ease of interpretation [106]. A variety of mass spectrometric techniques, developed based on this method, can detect very low elemental pressures (>2 × 10^−13^ pa of permeant) and simultaneously control transport parameters (e.g., flow rate, humidity of the gas, gas pressure) [107]. In gas permeation studies (Table 3), mass transport parameters of epoxy materials are usually obtained using dry gas mixtures to prevent sample moisture retention (and associated confounding effects). Diffusion cells consist of a supplier chamber containing the permeant feed gas, a receiver volume to continuously monitor the amount of permeant in an inert carrier gas, and a test specimen serving as a barrier between chambers (Figure 3) [108]. Although the data based on such experimental techniques can provide a range for the mass transfer properties of the epoxy, they do not reflect the consequences of interactions between water and other permeants in wet diffusion systems. For instance, comparing mass transport data of water and oxygen from Table 1, Table 2 and Table 3 shows that water transmission through epoxy is significantly higher than oxygen (i.e., >10^−13^), which has a higher diffusivity in epoxy. Still, there is an absence of literature data to explain the transport in a water containing system.

**Table 3 polymers-13-01517-t003:** Oxygen mass transport parameters (dry conditions) in epoxy polymeric structures from the literature data.

Material	Test Temp.	*T_g_*	*l*	*P* × 10^16^	*D* × 10^13^	*S* × 10^5^	Ref.
	(°C)	(°C)	(µm)	(mol/m−s−Pa)	(m^2^/s)	(mol/m^3^−Pa)	
DGEBA/PGE/DDM	30	117	20	-	1.60	20.9	[109]
DGEBA/DDM	30	164	20	-	2.40	18.0	[109]
	40	164	20	-	2.80	30.3	[109]
Epon828/A2049	25	150	~150	0.47	4.97	9.44	[110]
	40	150	~150	0.80	10.1	7.63	[110]
	65	150	~150	1.70	32.8	5.19	[110]
	80	150	~150	2.57	52.2	4.82	[110]

Gas transport through glassy polymers under dry conditions has been widely explored for gas separation applications, where the focus is on permeability changes according to free volume variations in polymeric membranes [112,113]. It was found that the correlation between the gas permeability coefficient and free volume size in polymeric membranes contributes more to diffusivity than solubility [114]. Counter-intuitively, non-destructive Positron Annihilation Lifetime Spectroscopy (PALS), shows that gas solubility does not change with free volume variation in a glassy polymer; instead, sorption is more affected by factors such as *T_g_* and interactions between the penetrant, additives, and the polymer [115]. Using PALS to conduct free volume measurements, a model was built based on tortuosity variations associated with different amounts of nanoclay additives, which suggested a logarithmic-inverse relationship between diffusivity and cavity free volume (e.g., 10 times less diffusivity with a 15% smaller free volume size) [116]. Nonetheless, one must always note the effect of temperature on the stability of the polymeric network for a mass transport analysis. For example, at temperatures close to *T_g_*, permeability of gases with low chemical affinity for the polymer (e.g., oxygen) experiences a transition, which originates from a solubility change due to the formation of larger holes at higher temperatures [110]. This means that, contrary to prior investigations [114,117], diffusivity is associated with the path length between free volume sites, not the free volume size.

When a polymeric solution is cooled down in a practical time scale, the cooling rate usually exceeds the relaxation rate of polymer chains and the resulting solid contains more free volume space than is expected at equilibrium conditions [118]. According to free volume theory [119], a polymer volume consists of three elements (Figure 4):occupied “van der Waals” volume which is not a function of temperature,interstitial free volume stemming from vibrational energy of polymer bonds which increases marginally with temperature; andhole free volume which is related to volume relaxation and plasticization upon heating and cooling of the polymer.

The last element is the accessible volume for permeant transport and is substantially governed by temperature. In glassy polymers, an excess free volume (illustrated as *V_g_ − V_l_* in Figure 4) is trapped as a result of slow relaxation and since absorption and desorption of penetrant affect this volume, it is continuously redistributed through the polymer volume [120]. The extent to which this non-equilibrium excess free volume can increase is closely correlated with *T_g_*: the higher the *T_g_* is, the more the excess free volume is included in the polymer. When there is a strong interaction between a permeant such as water and a hydrophilic polymer, the non-equilibrium glassy state forces the polymer to gradually sorb more permeant to relax. When water is the penetrant, this sorption then lowers the mobility of freezable and non-freezable bound water inside the polymeric membrane [121]. According to the free volume theory [119], two mechanisms are generally involved in sorption and diffusion through glassy polymers: solution of low molecular weight species utilizing the equilibrium free volume in the dense portion of the polymer (Henry’s type sorption) and an additional “hole filling” adsorption associated with the excess free volume (Langmuir type sorption). The combination of these mechanisms is referred to as the dual-mode sorption (DMS) model. A conceptual illustration of sorption mechanisms including DMS is shown in Figure 5. According to the DMS model, the total amount of permeant (*C*) in the polymer for a binary diffusion system is a function of the Henry’s law dissolution term (*C_D_*) and a Langmuir-type adsorption term (*C_H_*):(6)$$C=C_{D}+C_{H}=k_{D}p+\frac{C_{H}^{\prime }b^{*}p}{1+b^{*}p}$$
where *k_D_* is the Henry’s law constant (mol/m^3^-Pa), *p* is the permeant pressure (Pa), $C_{H}^{\prime }$ and $b^{*}$ are the Langmuir hole capacity parameter (mol/m^3^) and the affinity constant (Pa^−1^), respectively.

**Figure 4 polymers-13-01517-f004:**
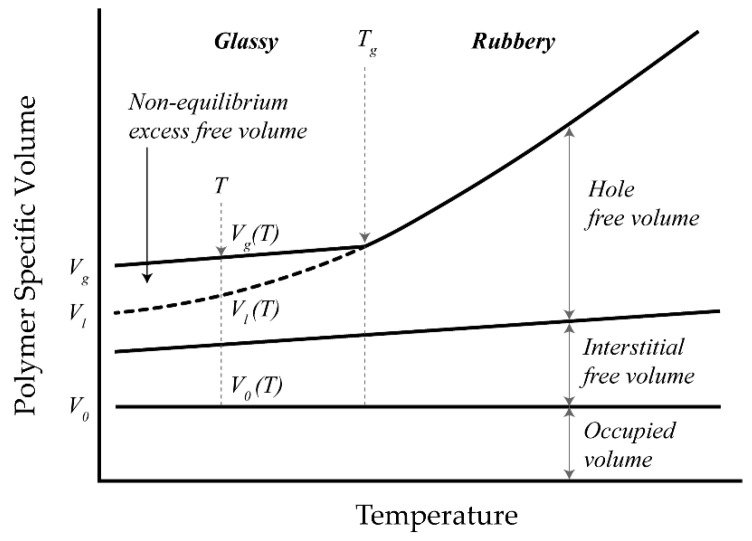
Schematic representation of polymer volume as a function of temperature. Reproduced with permission from [122]; *V_g_* and *V_l_* are the polymer specific volume (cm^3^) in the glassy and hypothetical rubbery state, respectively, and *V*_0_ is the occupied volume at 0 K.

The DMS model is used to study sorption kinetics of gas molecules in glassy polymers and to mathematically formulate the immobilizing effect associated with the adsorption process [124]. By modifying Fick’s second law, according to Equation (6), one can explain partial (or total) immobilization of gas species in the polymer, which then allows for the calculation of the concentration dependence of gas diffusivity and permeability [121]. Diffusivity is defined as the product of a kinetic mobility parameter, *L* (m^2^/s), and a thermodynamic parameter, τ, accounting for the concentration dependence of the chemical potential of the diffusing species [125,126]. The kinetic mobility parameter is usually attributed to diffusion of binary gas mixtures through glassy polymers: when the volume fraction of bound water in the polymer is highest, their mobility is the lowest, which partially immobilizes the non-polar oxygen gas and interrupts its adsorption at Langmuir sites [127,128,129,130]. On the other hand, gas solubility is expected to decrease when the adsorption sites in the polymer are saturated with another permeant: polymer hydration lowers the affinity constant for gas sorption (i.e., $b_{gas}^{*}$ in Equation (6)) [129]. It was shown that Langmuir sorption is the dominant factor in the total solubility of a gas and that more condensable gases have a higher solubility than less condensable ones—i.e., a higher magnitude of $C_{H}^{\prime }$ value in Equation (6) [123]. Table 4 summarizes important physical properties, including condensability (shown as ε/k), for water vapor and some gases of interest. Such data indicate that in wet-state conditions, water vapor sorbs into the Langmuir sites in the glassy epoxy in larger numbers compared to non-condensable oxygen, for example. Importantly, a low partial pressure of permeating oxygen, such as that consistent with pipeline operating conditions (i.e., atmospheric pressure surrounding the pipe), may not exert sufficient driving force to support the thermodynamic parameter (i.e., τ) for its diffusivity. Our preliminary measurements on free FBE films confirm that oxygen permeation rate undergoes a significant decrease in the presence of competing water from 25 to 45 °C [131]. It should be noted that antiplasticization responses from additives (in the FBE blend) may also affect the gas permeability by changing the water absorption process [132,133].

According to a well-developed model based on empirical data from gas separation membranes, there is a trade-off relationship between permeability of binary gas pairs and selectivity of the polymer for gas *i* over gas *j* ($\alpha_{ij}$). The so-called “upper bound” theory suggests that polymers that are more permeable are less selective and an upper bound to this relationship has the following mathematical form [113]:(7)$$\alpha_{ij}=P_{i}/P_{j}=\beta_{ij}/P_{i}{}^{\lambda_{ij}}$$
where *P_i_* is the permeability of the more permeable gas and *P_j_* is the permeability of the less permeable gas in light binary gas pairs. According to the following equations, $\lambda_{ij}$ is empirically correlated only with gas size and $\beta_{ij}$ (in units same as permeability) is a function of $\lambda_{ij}$ and a solubility selectivity term:(8)$$\lambda_{ij}={\left(\frac{d_{j}}{d_{i}}\right)}^{2}-1$$
(9)$$\beta_{ij}=\frac{S_{i}}{S_{j}}S_{i}{}^{\lambda_{ij}}exp\left[-\lambda_{ij}\left(b-f\frac{1-a}{RT}\right)\right]$$
where *d* and *S* are the kinetic gas diameter (m) and solubility (mol/Pa-m^3^) of gas *i* or *j*, respectively. In addition, *a* has a universal value of 0.64, b=11.5 cm^2^/s for glassy polymers, *f* is an adjustable parameter approximated to be 12,600 cal/mol. Modeling of penetrant solubility in polymers—developed from classical thermodynamics—correlates the sorption coefficient with condensability according to Equation (10):(10)$$lnS_{i}=M+N\left(T_{x}\right)$$
where $T_{x}$ is a condensability metric (e.g., critical temperature or ε/k in units of K), and *M* and *N* are empirical parameters and their average values are –6.91 cm^3^(STP)/(cc-cmHg) and 0.0153 K^–1^, respectively [136]. The latter approximations for the solubility coefficients are based on general trends of multiple sets of data found in the literature for light gas pairs (e.g., O_2_/N_2_, CO_2_/CH_4_, H_2_/N_2_). One can apply these values with the notion that these parameters are strongly affected by the gas pressure used in the experiment—they are susceptible to variations between polymers that show pressure dependence to gas sorption [136]. The upper bound line (drawn on a log-log plot of $\alpha_{ij}$ versus $P_{i}$) was found to be valid for gas pairs chosen from the common gases. However, the upper bound observation has also been proposed for systems where water is present in the permeate stream [137] and the gas permeation through a coating membrane in a diffusion cell is likely to follow similar mechanisms. In a coating performance study, such an analysis could provide an upper bound limit for mass transport when the coating system undergoes an adhesion failure (i.e., formation of a crack-free blister or a disbondment)—the permeant flux becomes the rate-determining parameter. While estimations of transport properties based upon this theory are in impressive agreement with empirical values for stiff-chain glassy polymers (i.e., amorphous polymers with high *T_g_*, rigid backbone, and relatively high fractional free volume), the theory does not account for the influence of permeant concentration on permeation properties—especially at temperatures away from the range of 25–35 ºC, which is a reference range for most of the above adjustable parameters. In other words, elevated temperatures move the polymeric membrane toward the rubbery state in which penetrants are weakly sieved based on size [113], thereby direct empirical estimations of selectivity would be more accurate. Furthermore, depending on the polymer and the gas mixture, a membrane that is selective for a condensable gas is likely to become more selective for less condensable gases at elevated temperatures (e.g., Figure 6) [138].

In addition to the sorption behavior characterized by DMS, clustering of water molecules inside the polymer network due to high water activity can generate a pooling effect [139]. The cluster formation, or self-association of water molecules, gives rise to a positive deviation from Henry’s law and affects the sorption and transport process in the polymer (Figure 5) [140]. This effect has been observed in epoxy structures as a result of the synergistic effects of high humidity (>50% RH) and temperature (>45 °C), which causes irreversible damage to the epoxy network [36,141,142]. An increasing population and accumulation of clusters at higher water activities decreases the diffusivity of water; this negative concentration dependence arises from the relative immobilization of water molecules upon clustering [143]. The opposing effects of cluster formation on solubility and diffusivity of water lead to uncertainty for its permeability coefficient (i.e., according to Equation (1)) [144]. On the other hand, the DMS model shows that $C_{H}^{\prime }$, which represents the maximum amount of permeant (i.e., water) sorbed into microvoids, decreases upon temperature increase [123]. The decline in transport properties of the water may result in higher gas diffusivity in the polymer. In other words, assuming the concentration of mobile water inside a water/polymer mixture controls the gas permeability in the system [144], one can expect a partial mobility of gaseous permeant upon increasing gas concentration at elevated temperatures—i.e., to support the thermodynamic parameter of gas diffusivity. To our knowledge, little direct experimental evidence exists on the effect of water clustering (and its temperature dependence) on water permeability and associated gaseous mass transport of epoxy coating membranes.

### 2.3. Water/Salt Transport

The water content of the polymer strongly affects the diffusion of salt in the polymeric membrane. In the solution-diffusion model for desalination applications, the volume fraction and sorption coefficient of water in the polymer are used interchangeably (i.e., $\nu_{W}\;$≈ $S_{W}$ in Equation (2)) and their units are [g(H_2_O)/cm^3^ (swollen polymer)]/[g(H_2_O)/cm^3^(solution)], where “cm^3^ swollen polymer” indicates the volume of the polymer together with any absorbed water and salt [60]. According to the free volume theory [119], competitive diffusion between hydrated ions through a polymeric network depends on their effective sizes. Table 5 presents size attributes for several ions of interest. Crystal radius (or an ion’s size in a crystal lattice) is defined as the distance between the ion’s nucleus and the outer-most electrons. Hydrated radii are calculated based on the Stokes-Einstein model and a calibration procedure that considers ion radii in aqueous solutions—i.e., the crystal radii of bulky, symmetric ions are set equal to their hydrated radii because they do not hydrate in water [145]. Due to variability in the extent of ion hydration in a polymer network, the crystal radius of the ion may commonly be assigned a lower limit on its effective size for diffusion in polymers [146]. However, for ions that do hydrate in water, the effective ion size is properly set equal to the hydrated radius and a general behavior similar to an aqueous solution can be expected for transport in polymers. For example, between two cations of the same valence (e.g., data for Na^+^ and K^+^ in Table 5), the ion with a larger crystal radius has a smaller hydrated radius because the charge density surrounding the larger cation is less than the smaller one. On the other hand, anions have less interaction with water molecules compared to cations and have more similar crystal and hydrated radii than that of cations (Table 5). Thus, for instance, despite having larger crystal radius than that of sodium, chloride can penetrate faster than sodium in the polymeric network because of its lower hydrated radius.

From the barrier properties perspective, permeability and selectivity of the membrane toward salt/water systems are commonly viewed as intrinsic transport properties of the membrane [146]. These measures are less dependent on boundary conditions during the permeation analysis than are permeance and salt flux, which are often reported by manufacturers [147]. In an ion permeability measurement, hydrostatic pressure and salt concentration gradient for transport are unidirectionally imposed across a membrane and using mass flux equations for salt and water, the membrane’s ability to separate two components is discussed [148,149]. The trade-off relationships developed for gas permeability systems (i.e., upper bound theory) are also observed for desalination membranes—i.e., polymers more permeable to water will naturally have less tendency to separate water from salt solutions [146]. The selectivity (or permeability selectivity), α, for water/salt separation is defined as follows:(11)$$\alpha_{W/S}=\frac{P_{W}}{P_{S}}=\frac{S_{W}}{S_{S}}\times \frac{D_{W}}{D_{S}}$$
where $S_{W}/S_{S}$ and $D_{W}/D_{S}$ are reported as water/salt solubility selectivity and diffusivity selectivity, respectively. Although data in the open literature to provide robust versions of tradeoff plots are still scant, it seems certain that most of the tradeoff relations between water permeability and water/salt selectivity come from changes in diffusivity—i.e., variations of permeability selectivity for different materials were based primarily on variations of diffusivity selectivity [147]. This sensitivity toward diffusivity selectivity is in accord with performance characteristics observed in light gas separations and qualitatively supports the fact that there are large differences between effective diameters of water and hydrated salt ions (cf., Table 4 and Table 5) [145,147].

Using free volume theory, Yasuda et al. [150] related the limiting diffusivity of a dissociated salt component ($D_{i}$) to the water content of the polymer network [150]:(12)$$D_{i}=D_{i}^{0}\times exp\left[b\left(1-\frac{1}{\nu_{W}}\right)\right]$$
where $D_{i}^{0}$ is the diffusion coefficient of the salt component *i* in pure water and *b* is an adjustable constant related to the ion size. Yasuda et al. [150] also found a linear dependence between average NaCl diffusion coefficients and 1/$\nu_{W}$ in a variety of polymers. This finding suggests that if the average diffusivity of dissociated salt components is much more sensitive to water content than solubility coefficient, their permeability will also correlate with the reciprocal of the water sorption coefficient (cf., Equation (1)). Local salt concentration (or external charge in the system) can also affect the kinetics of ion permeation [146]. Although ion size is a determining factor for salt mass transfer, electrostatic forces between diffusing ions and the polymer network are more dominant: they may inhibit salt sorption and lower the solubility of the salt from the limiting factor of water volume fraction in the hydrated polymer (i.e., *S_S_* < $\nu_{W}$) [148]. The presence of charge in the polymer can also influence the extent of ion hydration and effectively reduce the size of an hydrated ion—i.e., render them smaller than observed in dilute aqueous solutions (i.e., Table 5) [146].

Differential scanning calorimetry (DSC) can be used to determine the state of water and thus model the salt diffusivity as a function of $\nu_{W}$ (e.g., NaCl diffusion in [151]). Upon hydration, the polymer network separates into two phases: (1) the polymer and any non-freezable bound water and (2) free and freezable bound water. The non-freezable water molecules interact with hydrophilic groups in the polymer via hydrogen bonding or electrostatic forces and the freezable water governs dissolution of dissociated salt components. With respect to applications in soil or seawater, it is important to note that because of the slightly positive potential of epoxy resin coatings, Cl^−^ ions tend to penetrate the matrix and neutralize the existing charge [152]. However, time-dependent impedance measurements in sea water solution showed slow transport of Cl^−^ as compared to water because sodium diffusion is slower and rate determining [153].

## 3. The Long-Term Stability of FBE and Its Effects on Mass Transport

Adhesion is a key characteristic in coating degradation. Adhesion failure, and the resultant underlying corrosion, facilitate the flux of aggressive species across the coating. However, unless a major defect (e.g., microcracking) evolves from internal stresses in the coating membrane, barrier protection remains governed by transport processes through the coating. During the initial coating process, when a liquid FBE film flows onto a pretreated steel surface (i.e., cleaned with sandblasting and acid washed), hydroxyl groups along the molecular chain enable the coating structure to form robust anchor points. It is also necessary that the terminal epoxy groups adequately crosslink in the profile of the film to minimize internal stresses [154]. The resin molecule in FBE contains a three-membered ring, oxirane, which is highly reactive when curing takes place at high temperature—i.e., 180 °C to 200 °C for Low Application Temperature (LAT) FBE and up to 250 °C with standard and high *T_g_* FBE [155,156]. Apart from its processing advantages, this layer does not require the solvent to keep the binder and filler parts in a liquid suspension form. The final non-crystalline solid remains a glassy polymer at the operating temperature (e.g., 65 °C). Modifications of FBE molecular network (e.g., bromination of the phenyl functional group) result in *T_g_* increase, and consequently, improvements in adhesion and CD performance of the final coating [157]. Dispersion of filler materials such as carbon black in the powder blend can also enhance barrier properties as a result of decreasing porosity and chain segment motions [158]. However, one must note that a higher amount of filler particles increases the tendency for agglomeration, and thus, deteriorates barrier properties. In addition, the size and shape of the filler particles play important roles in erosion resistance [159,160,161].

In pipeline coating systems, myriad environmental and design parameters can exert a variety of forces to drive mass transport through the coating. In the early stages of degradation, Fickian diffusion takes place as a result of the water concentration gradient (i.e., vapor pressure difference or in the case of offshore areas, hydrostatic pressure) across the polymer coating. Likewise, the presence of salt in the vicinity of the coating (such as sodium and chloride in seawater or soil) results in transport of ionic species through the system. Measurements can become intricate when an applied potential is introduced to the system: the external electron source—e.g., cathodic protection (CP)—supports the ionic flow through electrostatic forces [146]. In addition, the presence of environmental CO_2_ (as in soil) makes the pipeline steel susceptible to stress corrosion cracking; this type of corrosion attack is most likely to occur at coating imperfection areas [162]. However, the high tendency of condensable gases like CO_2_ to sorb into microvoids of glassy polymeric coatings can facilitate their gradual gas permeation [123,163]. Relevant data such as CO_2_ transport through coatings, especially for hydrated epoxy, are missing in the literature.

The FBE coating may begin to degrade prior to the service life of the pipe due to storage conditions, for example, via ultraviolet (UV) exposure during stockpiling in the field [164]. Physical ageing of glassy polymers leads to rapid deterioration of transport properties, usually accompanied by an increase in selectivity of the polymer against a gas mixture [114]. Studies on gas separation membranes have shown that thinner coating layers physically age more rapidly [165]; because the standard application thickness of FBE coatings on pipelines is thin (i.e., 350–500 µm), they fall into this accelerated ageing category. Latino et al. [92] found that hydrothermal ageing of FBE at 85 °C for three to seven months results in a minimum for *T_g_*. This then results in an increase of the electrical conductivity of the FBE, and despite further water uptake by the coating, *T_g_* does not subsequently drop. FTIR analysis of their FBE samples (Figure 7) showed that, in addition to O–H stretching (peaks between 3610 cm^−1^ and 3210 cm^−1^) from water uptake, new vibrations occur in other chemical bonds in the FBE structure (e.g., water reaction with carbon double-bond groups at 900 cm^−1^) [92]. On the other hand, they observed a signal increase at 1608 and 1732 cm^−1^ for aromatic rings and esters stretches, respectively, which may reflect chain scission of the epoxy backbone component and subsequent leaching of smaller molecular weight products towards the surface. This effect was previously reported for epoxy structures: at the maximum water sorption, the diffusion coefficient of oxygen decreases, suggesting chain scission and loss of volatile products in the epoxy. However, it has also been observed that this scission is followed by the formation of the new cross-links [109,166]. The *T_g_* of the coating may partially be recovered upon removal of water through high vacuum drying, which can result in limited reversible cross-linking [46].

### 3.1. Effect of Coating Thickness

The interaction of exposure environments with the coating is a function of the coated layer profile, thus a deficiency of thickness, as well as other production errors (e.g., poor surface preparation and improper curing), can result in corrosion of the underlying substrate earlier than the expected nominal service life of the coating system. FBE application procedures do not result in consistent coating thickness [167]; this adds a confounding variable for a systematic study. TGA analysis on epoxy polymers suggests a two-step mechanism for the material degradation: a fast surface degradation and a slower weight loss stage increasing with temperature but decreasing with thickness [109]. Above a minimum thickness, degradation of FBE coating is expected to be controlled by chain scission (leading to formation of volatile products) and migration of bulky species (e.g., carbonyl and amide) [109,166]. Site investigations of FBE-coated pipes also show that cathodic delamination and extent of mechanical damage to the coating intensify below a minimum thickness range (e.g., 350‒500 µm) [168,169]. In addition, the oxygen permeability coefficient through glassy polymers in dry and water-saturated conditions is inversely related to thickness [110,129,170]. This effect might be in part related to increase in the amount of excess free volume associated with thicker coatings [129]. It is important to note that, although increasing film thickness reduces the overall permeating flux, polymer oxidation at higher thicknesses can delay O_2_ ingress from the expected Fickian sorption process and lower the permeability over time [110,171].

### 3.2. Salt Deterioration

Inorganic salt contaminations due to poor substrate preparation can pull water into the coating faster and this is simply because of a decrease in vapor pressure if and when the permeating water makes a salt solution; the resulting osmotic pressure then becomes a driving force to cause blistering [172]. Water transmission induced by this blister growth is highly contingent on osmotic pressure. Some mathematical models have been suggested to predict osmotic pressure based on salt availability at the interface [173]. The highest osmotic pressure is initially formed when salt is at the saturation level in water. Upon dilution of salt with water, osmosis effects drop with time and the water movement across the coating slows. The osmosis effect is rooted in the water solubility of the particular salt. Hence, the relative threat for each salt is assessed based on their degree of dissociation, not the nature of the anion [172]. Table 6 shows water solubility of common salts and provides a solubility ranking in terms of the number of ions in solution at saturation. It suggests that, for instance, ferrous chloride puts more ions into solution at saturation than does sodium chloride, and so FeCl_2_ is a bigger osmotic threat than NaCl. In addition, dissociation of most salts is a function of temperature which can shift their osmotic pressure (Table 7). Interestingly, when compared to many other salts in Table 7, NaCl has the lowest variability of water solubility with temperature. This may explain why salt spray tests based solely on NaCl (5 wt% NaCl fog at 35 °C as per ASTM B117) are unreliable accelerated corrosion tests to represent industrial atmospheres [174]. Including other species such as sulfate and ammonium as well as sodium chloride (e.g., ASTM D5894) improves the correlation of exposure tests with practical environments [175,176]. Such a test protocol allows objective assessment of coating/metal systems and enables quantitative measurement of coating properties after exposure—e.g., using EIS analysis to study the decrease of bulk-coating layer barrier properties due to exterior exposure [177].

Although most documented blistering failures are related to the presence of salt contaminants prior to coating application [6,168], some reports show ion transport (e.g., NaCl) also takes place through FBE [180]. A study on cathodic delamination of various epoxy coatings indicates that resistivity of the coating toward ion transport increases with thickness [181]. This direct relationship was attributed to the formation of low resistance areas (i.e., regions of low crosslink density or low molecular weight) which are distributed across the coating profile; increase of thickness may partially eliminate this heterogeneity. Wei et al. [182] reported an accelerated transport of NaCl solution through an FBE coating under flowing conditions as compared to static immersion: in stagnant solutions, electroneutral forces from accumulated positive charges on the coating surface slow ionic movement, whereas in flowing conditions, these forces are removed and ions penetrate faster into the coating. Such a finding suggests that exposure to a flowing corrosive solution may have little effect on water diffusion, but can progressively deteriorate the epoxy network by facilitating ion transport.

Only a few studies have quantified the amount of salt that might diffuse through an FBE coating and some of these are not representative of industrial conditions. For example, the cross-section profile of a blistered FBE coating showed no indication of sodium diffusion after a 48-h test [168]; such a short-term test cannot generate solid information on ion transport. The slightly smaller effective size of Cl^−^ ions compared to that of Na^+^ (cf., Table 4) may also relate to its faster penetration into the macropores of the polymer network [183]. Another study on ionic transfer resistance of epoxy coatings (~30µm thick) suggests that at salt concentrations above 3.5 wt.%, salt crystals tend to form inside the coating (after 30 days), which can block the micropores and decrease the rate of ion penetration [103].

### 3.3. Effect of CP on Ion Transfer

Direct current (DC) interference may increase water permeation and ionic diffusion in FBE coatings as was indicated through IR spectra—some adsorption peaks were observed only when DC voltage was present [184]. Kuang and Cheng [185] proposed a three-step model for water uptake in the presence of CP for FBE-coated steel:a monolayer of water covers the coating layer and binds to exposed polar groups;water molecules jump to existing micropores in the cross-linked FBE structure, destabilize ether (C–O–C) and hydroxyl functional groups, disrupt hydrogen bonds, and create more pores;water and ions permeate through the micropore channels as a result of electroosmosis caused by CP potential.

Although this model provides an explanation for mass transport through the FBE structure, analytical studies are required to validate it and to quantify the extent of ion permeation through the coating profile. To protect the underlying substrate in the disbonded area, CP application must result in ionic current through the disbonded coating. However, this current also contributes to hydroxyl ion generation, which increases the alkalinity of the solution surrounding the cathode as follows:(13)$$O_{2}+2H_{2}O+4e^{-}↔4OH^{-}$$
(14)$$2H_{2}O+2e^{-}↔H_{2}+2OH^{-}$$

Latino et al. [186] simulated the environment under disbonded pipeline coatings (i.e., FBE and HPPC) with a Ti working electrode and a pH electrode at low CP (–850 mV vs. Cu/CuSO_4_) and at room temperature. The instantaneous current and pH showed that both inward migration of protons and outward migration of hydroxyl ions from the disbonded area did not significantly affect the local pH after continuous CP application. Using a high sensitivity potentiostat plus a Faraday cage to capture small currents (pA/cm^2^), these workers showed that the low CP and temperature did not produce alkalinization in the disbonded area. This finding indicates that ionic current transport through intact FBE is insufficient to protect the steel cathode i.e., it remains too resistive. Importantly, they argued against the generally accepted “non-shielding” properties of FBE and related the effective CP protection of FBE coatings to probable pinholes and microcracks that develop because of water uptake and ageing. More recent work from Latino et al. [92] confirmed that CP permeability was significantly higher in aged than unaged FBE, implying again the ageing-induced pinholes and microcracks must be present to confer adequate ionic transport through the coating. They also showed that diffusion driven by steep hydroxyl ion and proton concentration gradients across the film generally predominates over electroosmosis due to the CP potential gradient. Minor effects of cathodic polarization on ionic transport were also reported by Sørensen et al. [181], who also found an improvement in the resistance toward cathodic delamination with increasing concentration of secondary hydroxyl groups in the uncured epoxy resin (after 18 weeks of exposure to KCl solution at 18 °C). They related this effect to an increased number of interaction points between the coating and substrate surface (not to decreased permeability toward ions, gases, and liquids).

The above discussion strongly suggests that, without a defect to allow ionic conduction, an external CP potential may only very weakly polarize the steel under a disbondment area. Qian and Cheng showed that localized direct current interference (i.e., up to 20 V) can generate cathodic and anodic zones on the coating surface that promote water transport (and consequently ionic transport) in the polymeric network [184]. However, morphological characterization of the tested FBE coating confirmed the development of microcracking in the coating structure, which might be the actual cause of the resulting cathodic and anodic zones. Thus, it should not be a great surprise that CP penetration is mostly an effective parameter when a holiday exists around a CD area. In such a case, the effectiveness of CP to mitigate corrosion is highly dependent upon the potential difference between the holiday area and the inside of the disbonded area [37,187]. Adequate CP usually results in the formation of an alkaline environment at the holiday (e.g., pH reaching 12 at −1.126 V vs. Ag/AgCl). Due to the limited diffusion through the disbonded crevice (or IR drop in the narrow gap) an adverse side effect of adhesion loss (i.e., CD) might occur through simultaneous protection by coatings and CP [188]. Accordingly, application of CD test methods is necessary to monitor the disbondment behavior of the in-service coatings and to achieve the proper reference for adjustment of CP potential over time [188].

## 4. Multi-Layer Coating Systems

FBE has low resistance to abrasive stresses and significant damage to FBE-coated pipes occurs after brief exposure to mechanical stresses [32]. With increased coating system requirements such as high temperature use, and improved resistance to abrasion and chemicals, three-layer polyolefins (3LPO) systems become the preferred choice [1]. A 3LPO coating typically consists of an FBE primer, a polyolefin adhesive, and a polyolefin topcoat—mainly high-density polyethylene (HDPE) or polypropylene (PP). Each successive layer is designed to adhere to the next by gradually altering the chemistry to aid compatibility and chemical interaction from the steel pipe surface to the outer layer topcoat [189,190]. The multilayered approach requires minimum interfacial stresses between layers to ensure the final coating can achieve optimal adhesion and barrier properties. Design parameters such as the thickness of each layer and application temperature are thus of equal importance to the layer compositions [1]. Finite element modelling of a 3LPO coating showed that increasing the polyethylene (PE) topcoat layer thickness reduces the value of stress at the PE/epoxy interface but increases the stress at the epoxy/steel interface [191]. Processing of polymer structures at temperatures high above their *T_g_*s can result in mutual interdiffusion of two distinct polymers across an interface. For example, a diffusive interphase layer of 10–1000 Å leads to strong entanglements between two compatible polymers [192].

Another multi-component system is HPPC. It is expected to have low susceptibility to internal stress development over wide temperature variations. It typically consists of a 175–250 µm FBE primer, a 125–150 µm polyolefin powder adhesive, and a 500–800 µm topcoat HDPE, all applied by electrostatic spray on the heated pipeline [31]. The desired thickness for each layer is selected according to project specifications and performance requirements—e.g., thicker topcoat HDPE profiles at areas susceptible to mechanical damage, weathering, or high humidity. This coating system benefits from the directional solidification of the polymer, which protects the coating from internal stresses and microvoids [31]. Mass transport studies on HPPC generally focus on coating performance in CD tests [7,186]. Permeability data for this coating at higher temperatures are limited and have been measured qualitatively, mostly relying on comparison with other coating structures [26]. Thorough investigation of mass transport in multilayered coatings requires empirical data on the barrier properties of PE components.

### 4.1. Mass Transport through the PE Barrier

Gas transport through PE structures has been extensively researched in the food packaging industry due to its impact on food shelf-life [193,194,195] (Table 8). The crystallinity of thermoplastics reduces the free volume in the polymer structure and increases the misorientation and distribution (tortuosity) of diffusion paths for permeants [196]. PALS experiments on semi-crystalline polymers have shown that the tortuosity of crystallites is more important than void size for gas permeation [197]; tortuosity adds a possible percolation element to the diffusion process and restricts the permeant mobility. Mrkić et al. [198] measured permeability of O_2_, CO_2_, N_2_, and air through PE laminates and correlated transport coefficient data at different temperatures with the physical characteristics of the permeants. They concluded that the gas permeability of PE increases at higher temperatures (50–60 °C) due to a reduction in crystallinity. They also found that adding polar amide groups to the PE structure decreases the permeance of nonpolar gases (O_2_, CO_2_, and N_2_) through amorphous regions. However, such a modification reduces the barrier properties of the polymer against polar permeants like water vapor [199,200]. Therefore, treatments to improve hydrophobicity and crystallinity of the coating bulk phase are preferred to prevent wet-state diffusion [201,202].

Different permeability values (reported in the literature in Table 8) can be due to microstructural variations among the tested materials. There are, however, discrepancies regarding the kinetics of gas permeation for the same nominal material (i.e., data for O_2_ and CO_2_ in [205]). According to the kinetic diameters of O_2_ and CO_2_ molecules (cf., Table 4), a higher permeability to CO_2_ is expected for the same accessible pore volume. High condensability of CO_2_ also results in a high solubility for this gas in the PE matrix, which leads to its higher permeability than O_2_—i.e., ${\left(\varepsilon /k\right)}_{{CO}_{2}}>{\left(\varepsilon /k\right)}_{O_{2}}$ according to Table 4 [196].

### 4.2. Analysis of Transport for Multi-Layered Membranes

Permeant transmission through a defect-free laminate composed of different polymeric layers essentially follows the ideal laminate theory [206]. Thus, for instance, gas permeation results for FBE and PE can yield a dependable estimate for permeation through HPPC. The permeability coefficient for a multilayer membrane, $P^{ml}$, is a function of the thickness ($l^{n}$) and permeability ($P^{n})$ of the constituting layers:(15)$$\frac{P^{ml}}{\sum^{\;}l^{n}}=\frac{1}{\sum^{\;}l^{n}/P^{n}}$$

However, due to unclear contribution of the constituting layers to the mass transfer resistance of multilayered membranes, permeance (*Q_i_* in mol/m^2^-s-Pa) is used instead of permeability to assess the membrane performance from direct empirical results:(16)$$Q_{i}=\frac{J_{i}^{ml}}{A∆p}=\frac{P_{i}^{ml}}{\sum^{\;}l^{n}}$$
where $J_{i}^{ml}$ and ∆p are the flux and pressure difference across the multilayered membrane, respectively, in permeation equations (e.g., Equation (4)). Unlike permeability, the permeance is heavily dependent on the thickness of the membrane and operation conditions (e.g., upstream and downstream pressure), yet comparison of permeance data from different tests can provide a solid database for selectivity of multilayered membranes [207,208]. Accordingly, a separation factor, $\alpha_{ij}^{*}$, of gas *i* over gas *j* is defined by [209]:(17)$$\alpha_{ij}^{*}=\frac{Q_{i}}{Q_{j}}$$

Although the separation factor is commonly used instead of the selectivity for composite membranes, one must note that the permeance and separation factor are sensitive to the test conditions [209,210], and the separation factor approaches the ideal selectivity only when significant gradients exist between up- and down-stream pressures [114]. Accordingly, to analyze barrier performance of coatings of interest, the effect of test parameters (e.g., temperature and pressure) can be studied within the classification of their standard properties—i.e., $\alpha_{ij}$ for FBE and $\alpha_{ij}^{*}$ for HPPC.

## 5. Coating Imperfections and Remaining Life Assessments

The literature concerning the time-dependent barrier performance of an a priori defect-free coating system is relatively inadequate. In other words, assuming that a near-perfect coating system is placed on a well-prepared steel substrate, one is not likely to find studies addressing questions such as: How long could a pipeline operator reasonably count on the coating retaining its initial, “ideal”, barrier properties? What effect might cathodic protection have on the deterioration of these properties, among other considerations? A modelling scheme to answer such questions requires performance assessments of the coating during consecutive steps of degradation. It also needs to connect mass transfer to the degradation process and predict the frequency of coating failures. To accomplish such objectives, possible modes of failure, as well as experimental approaches to generate attributes for probability of such events, need to be identified for a coating system of interest. For example, introduction of new interfaces between coating layers through the multilayering approach may increase the probability of developing defects, i.e., porosity in forming layers and insufficient adhesion between layers in the coating profile. Although effective pipe inspections minimize the occurrence of defects for in-service pipelines, progressive degradation can induce imperfections in the coating structure. An exemplar of such cases is sequential water absorption/desorption, which can develop internal stresses in the coating profile. This eventually leads to adhesion loss or microcracks in the final stage of degradation [211,212]. UV exposure also causes loss of mechanical properties and thereby lowers the durability of the coating [213]. Complementary mechanical properties tests such as adhesion pull-off and nanoindentation techniques can quantitatively show this decline [214,215]. In nanoindentation tests, for instance, the resistance of the coating layer against indentation force can be used as a proxy for the extent of degradation of its mechanical properties.

Predicting the remaining life of pipelines using numerical assessments has received attention in recent research [216,217,218]. The modelling scheme to predict the failure point can follow either deterministic or probabilistic approaches [219,220]. However, it is generally acknowledged that assessments for an existing structure referencing back to design specifications (as in deterministic methods) for such complicated systems are likely to result in excessively conservative estimates of remaining life [221]; they are likely to require more structural capacity than is reasonably necessary to fulfill both safety and performance criteria. Common modes of failure for each coating type have been documented in some existing reports [168,222].

This review addresses the consequences of exposing a coating system to challenging environments, however, diverse mechanisms of coating degradation indicate that one cannot directly use the transport data, if known, to formulate a model for the overall degradation process. Alternatively, probabilistic methods to relate likelihood of failures based on the declining properties of the coating appear to be a good fit [223,224]. Monte Carlo simulation is a widely used technique for risk assessment and reliability analyses in disparate engineering fields and is successfully used for lifetime predictions in oil and gas pipelines (i.e., to process corrosion wastage after coating failure) [225,226,227]. A modeling framework to address individual failure types based on mass transport analyses and mechanistic attributes of degradation (i.e., corrosion attack or disbondment) can generate signals for an ultimate failure (e.g., exposed pipe). In theory, a failure probability is determined by the variability of environmental parameters (e.g., history of humidity, temperature, UV exposure, etc.) and their interactions with mass transport properties of the coatings. Independent statistical distributions based on the empirical data generate a cumulative distribution function to predict the failure behavior of the coating [228,229]. Numerical solutions of the resulting evolution equation (due to gradual degradation) can be achieved by employing computational algorithms [230,231]. Such a numerical study is missing in coating degradation literature and would be helpful to reach optimal inspection intervals for oil and gas pipelines.

## 6. Summary and Future Outlook

Although water absorption into a polymeric coating may not be the rate-determining step in corrosion of an underlying pipe surface, it can gradually change the molecular structure of the polymeric coating and affect transmission of other species. Studies on water ingress are thus anticipated to shed light on gaseous and ionic transport properties of polymeric coatings and the mechanism of their degradation. From this may also evolve the notion that service parameters (e.g., temperature, humidity, or gas pressure) can affect polymer hydration and thus alter operative sorption mechanisms in the coating membrane. Some concluding remarks concerning the permeability of pipeline coatings toward wet-state environments are as follows:
Qualitative studies of water transmission rate at room temperature may provide a baseline to compare the extent of permeation among coating systems. However, they do not provide adequate information to assess coating performance, especially at elevated temperatures. Quantitative measurements of water transport based on its concentration and temperature are necessary and research in this area should be accelerated.Spectroscopic and electrochemical techniques yield a fundamental understanding of barrier properties of polymer structures. However, these methods may not be applicable for multilayered coatings. Properties of individual layers need to be related to the mass transport capacity of the composite coating.Although FBE shows high permeability to water, effects of water saturation on gaseous permeation have been poorly covered in the literature. Water is expected to have a partial immobilization effect on gas transport, which is sensitive to gas concentration and working temperature.Stockpiling coated pipes prior to their service life is a common practice by industry. Combined with moisture uptake, UV exposure can significantly affect the barrier properties of coatings. Analysis of UV exposure effects on the mass transfer capacity of these materials is lacking and is a requirement for corrosion protection assessment.Wet-state use can change mass transfer properties of polymers, depending on their molecular structure, in different ways than dry state use. Therefore, analysis from a corrosion model based on data from dry conditions may not generate an accurate assessment for wet-state conditions.A probabilistic model to predict the frequency and the extent of polymer degradation can increase the efficiency of pipeline inspection and corrosion protection.

## Figures and Tables

**Figure 1 polymers-13-01517-f001:**
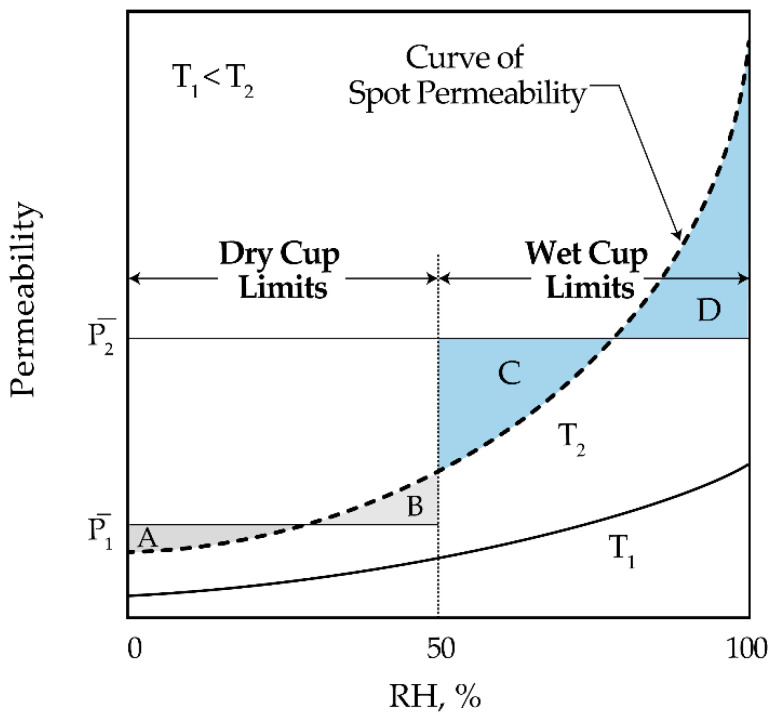
Variability of the spot permeability with water vapor concentration for WVT tests.

**Figure 2 polymers-13-01517-f002:**
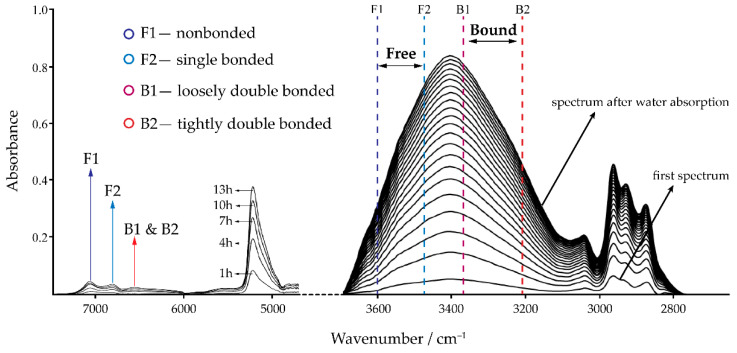
Water absorption peaks in near- and mid-range IR spectra; relative to associated bonding forces water molecules can affect the epoxy network structure. Copyright 2010. Reproduced from [89] with permission from Sage Publications Inc.

**Figure 3 polymers-13-01517-f003:**
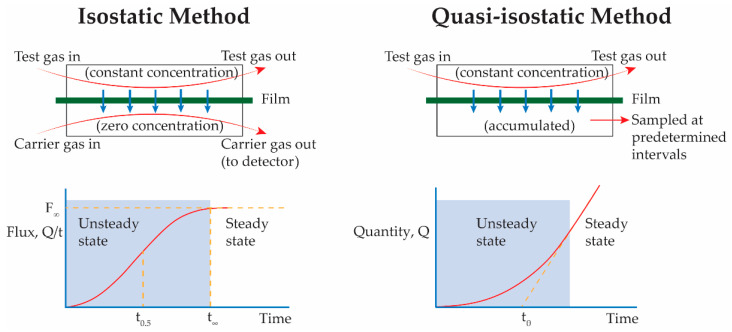
Schematic diagram and plot of common gas permeation measurement methods. In the quasi-isostatic (also known as time-lag) method, the downstream chamber is maintained in concentrations below 5 wt% of the upstream side throughout the test. Copyright 2018. Reproduced from [111] with permission from Elsevier Ltd.

**Figure 5 polymers-13-01517-f005:**
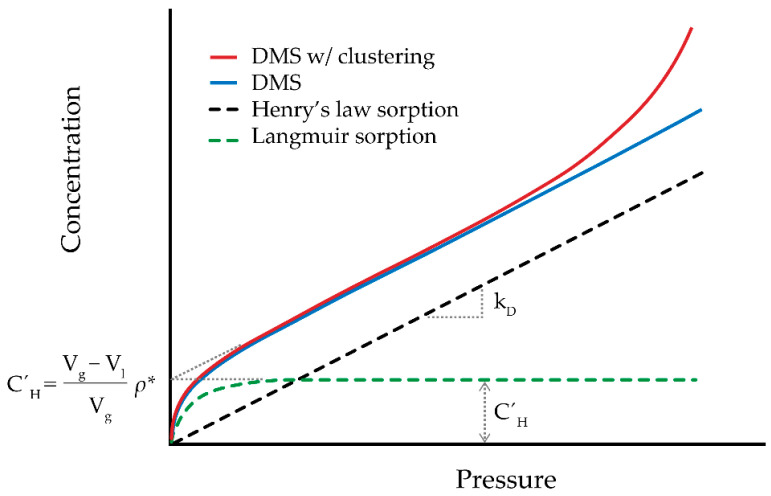
Schematic representation of permeant sorption mechanisms in polymeric structure. Mass density of permeant, denoted by $\rho^{*}$, is reciprocal value of its partial molar volume (mol/cm^3^). Reproduced with permission from [123]. *H* and *D* represent “holes” and “dissolved”, respectively.

**Figure 6 polymers-13-01517-f006:**
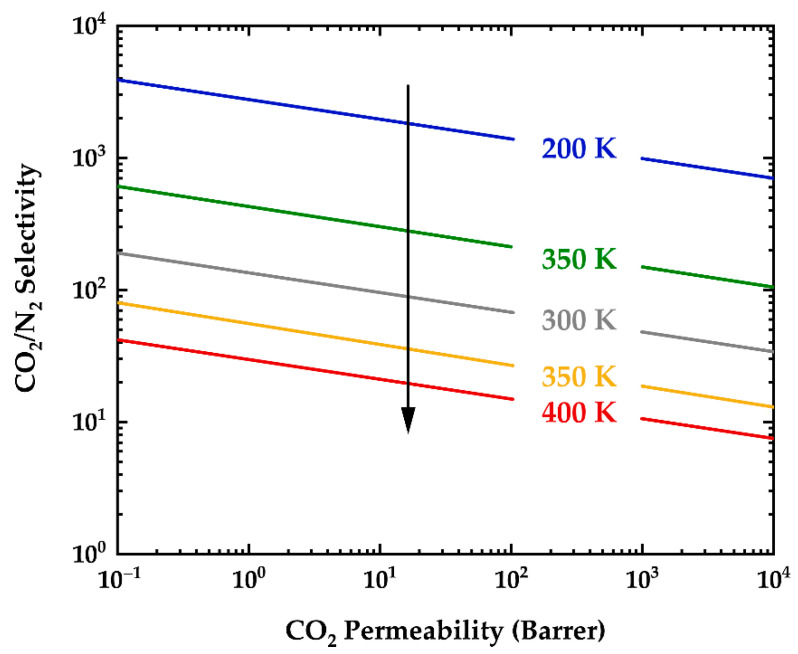
Influence of temperature on the upper bound selectivity for CO_2_/N_2_ separations. Copyright 2010. Reproduced from [138] with permission from Elsevier Ltd.

**Figure 7 polymers-13-01517-f007:**
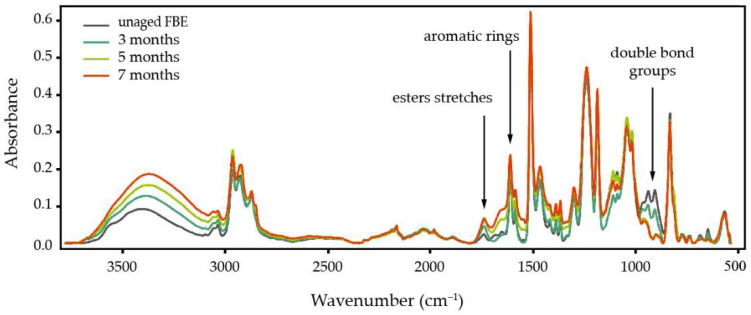
Degradation of FBE molecular structure caused by water sorption: after three months hydrothermal ageing, water can disrupt epoxy double bonds and cause chain scission for the backbone. Copyright 2019. Reproduced from [92] with permission from Elsevier Ltd.

**Table 1 polymers-13-01517-t001:** Water vapor diffusion coefficients in different epoxy compositions based on gravimetric uptake measurements.

Material	Test Temp. (°C)	*T_g_* (°C)	*l* (µm)	*D* × 10^13^ (m^2^/s)	Ref.
DGEBA/DBP	20	120	1500	1.77 ^a^	[62]
	40	120	1500	11.4 ^a^	[62]
DGEBA + mPDA	45	173	1150	3.35	[37,38]
	60	173	1150	7.93	[37,38]
EPON 828RS	22	80	~245	0.53	[63]
	60	80	~245	13.6	[63]
FBE	60	~110	500	3.00	[27]
DER332 (Semirigid 80%)	35	146	300	3.41	[42]
	70	146	300	13.5	[42]
DER332 (Flexible 100%)	35	113	300	1.61	[42]
	70	113	300	14.4	[42]

^a^ Resulted from only the equilibrium water uptake measurements, excluding the associated non-Fickian uptake.

**Table 2 polymers-13-01517-t002:** Water vapor permeance (*P/l*) and permeability (*P*) data from cup methods.

Material	Test Temp.	RH	*T_g_*	*l*	*P/l* × 10^9^	*P* × 10^13^	Ref.
	(°C)	f/p (%)	(°C)	(µm)	(mol/m^2^−s−Pa)	(mol/m−s−Pa)	
Liquid epoxy ^a^	38	100/90	91	30	3.3	-	[73]
DER-SAA-1.0	37.8	100/0	158	100	-	~7	[72]
Epon828/A2049	65	100/0	150	78	-	2.6	[35]
Epon828/D230	65	100/0	150	83	-	3.8	[35]

^a^ With no additive particles.

**Table 4 polymers-13-01517-t004:** Physical properties of water and common gases.

Gas	Molecular Weight M (g/mol)	Critical Temperature ^a^ *T_c_* (K)	Kinetic Diameter ^b^ σ	Condensability ^a^ ε/k (K)
H_2_O	18.015	647.3	2.65	809.1
O_2_	31.999	154.6	3.46	106.7
CO_2_	44.010	304.1	3.30	195.2
H_2_	2.016	32.9	2.89	59.7
N_2_	28.013	126.2	3.64	71.4
CH_4_	16.043	190.4	3.80	148.6

^a^ Ref [134], ^b^ Ref [135].

**Table 5 polymers-13-01517-t005:** Measures of ion size. Copyright 2014. Adapted from [146] with permission from Elsevier Ltd.

Ion		Crystal Radius (Å)	Hydrated Radius (Å)
Iron (II)	Fe^2+^	0.75	4.28
Iron (III)	Fe^3+^	0.60	4.57
Magnesium	Mg^2+^	0.65	4.28
Sodium	Na^+^	0.95	3.58
Calcium	Ca^2+^	0.99	4.12
Potassium	K^+^	1.33	3.31
Ammonium	NH_4_^+^	1.48	3.31
Chloride	Cl^−^	1.81	3.32
Nitrate	NO_3_^−^	2.64	3.35
Carbonate	CO_3_^2−^	2.66	3.94
Sulfate	SO_4_^2−^	2.90	3.79

**Table 6 polymers-13-01517-t006:** Water solubility of common salts at 20 °C.

Salt	Formula Weight (g/moles)	Solubility (g/100 mL aq.)	Moles Ion	Total Ions (Moles)	Rank
FeCl_3_	162.21	91.8 ^a^	0.5659	2.264	1
MgCl_2_	95.23	54.6 ^b^	0.5733	1.720	2
(NH_4_)_2_SO_4_	132.14	75.4 ^a^	0.5706	1.711	3
FeCl_2_	126.75	62.5 ^b^	0.4930	1.479	4
NaCl	58.45	35.9 ^a^	0.6142	1.228	5
CaCl_2_·6H_2_O	219.07	74.5 ^b^	0.3400	1.020	6
KCl	74.55	34.2 ^b^	0.4587	0.9175	7
Na_2_SO_4_	142.04	19.5 ^b^	0.1372	0.4118	8
MgSO_4_·6H_2_O	228.45	44.5 ^a^	0.1947	0.3895	9
FeSO_4_·7H_2_O	278.08	48.0 ^b^	0.1726	0.3452	10

^a^ Ref [178], ^b^ Ref [179].

**Table 7 polymers-13-01517-t007:** Solubility of common inorganic salts in water at various temperatures (g/100 mL aq.).

Salt	0 °C	10 °C	20 °C	30 °C	40 °C	50 °C	60 °C	70 °C	80 °C	90 °C	100 °C
FeCl_3_ ^a^	74.4	81.9	91.8	-	-	315.1	-	-	525.8	-	535.7
MgCl_2_ ^b^	59.2	53.6	54.6	55.8	57.5	-	61.0	-	66.1	69.5	73.3
(NH_4_)_2_SO_4_ ^a^	70.6	73	75.4	78.0	81.0	-	88.0	-	95.3	-	103.3
FeCl_2_ ^b^	49.7	59.0	62.5	66.7	70	-	78.3	-	88.7	92.3	94.9
NaCl ^a^	35.7	35.7	35.9	36.1	36.4	36.7	37.0	37.5	37.9	38.5	39.0
CaCl_2_·6H_2_O ^b^	59.5	64.7	74.5	100	128	-	137	-	147	154	159
KCl ^b^	28	31.2	34.2	37.2	40.1	-	45.8	-	51.3	53.9	56.3
Na_2_SO_4_ ^b^	4.9	9.1	19.5	40.8	48.8	-	45.3	-	43.7	42.7	42.5
MgSO_4_·6H_2_O ^a^	40.8	42.2	44.5	45.3	-	50.4	53.5	59.5	64.2	69.0	74.0
FeSO_4_·7H_2_O ^b^	28.8	40.0	48.0	60.0	73.3	-	100.7	-	79.9	68.3	57.8

^a^ Ref [178], ^b^ Ref [179].

**Table 8 polymers-13-01517-t008:** Gas transport data for HDPE from the literature data.

Material	Temperature	*P* × 10^16^	*D* × 10^11^	*S* × 10^5^	Ref.
	(°C)	(mol/m−s−Pa)	(m^2^/s)	(mol/m^3^−Pa)	
	H_2_O				
HDPE (0.938 g/cm^3^)	23	83.7	-	-	[40]
HDPE (0.96 g/cm^3^)	25	40	-	-	[203]
	O_2_				
HDPE	23	2.2	-	-	[41]
HDPE (0.945 g/cm^3^)	25	3.39	1.96	1.76	[204]
HDPE (0.964 g/cm^3^)	25	1.33	1.70	0.80	[205]
	40	2.62	3.46	0.78	[205]
	60	5.89	8.10	0.75	[205]
HDPE	30	6.53	2.63	-	[196]
	40	14.6	4.91	-	[196]
	60	33.3	10.3	-	[196]
	CO_2_				
HDPE	23	6.01	-	-	[41]
HDPE (0.945 g/cm^3^)	25	10.4	1.48	9.64	[204]
HDPE (0.964 g/cm^3^)	25	1.2	1.20	0.98	[205]
	40	2.14	2.39	0.88	[205]
	60	4.29	5.43	0.88	[205]
HDPE	30	35.5	2.16	-	[196]
	40	52.9	3.42	-	[196]
	60	105	6.35	-	[196]

## Data Availability

The data presented in this study are available on request from the corresponding author.

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
