# Peer review of "A Critical Review of the Time-Dependent Performance of Polymeric Pipeline Coatings: Focus on Hydration of Epoxy-Based Coatings"

_polymers, 2021, doi:10.3390/polym13091517_

Round 1

Reviewer 1 Report

In this review work authors describe the time-dependent performance of epoxy-based coatings with a focus on hydration effects.  Analytical   techniques, including spectroscopic ones, are valuable tools for studying materials' behaviour in wet environment and for the evaluation of the long-term performance of epoxy coating systems.  As gas and ion permeability/selectivity strongly depend on level of hydration, it is of primary importance to study the transport effects governed by diffusion.  Authors discuss the role of service parameters, e.g. humidity, temperature, and concentration of aggressive species, and their role in transport effects.  The manuscript is well-written and based on 223 properly chosen References. However, some issues need to be further addressed:

  • in "Abstract: "... can dynamically develop new transport mechanisms" - what you mean by "new mechanisms" - are they not known in chemical engineering? "This review examines common analytical and spectroscopic techniques..." - are spectroscopic techniques not analytical?
  • it is a vast work - Table of Contents on the beginning would help the reader to get familiar with the scope; moreover, list of abbreviations is recommended,
  • anticorrosive functions in the epoxy coatings deserve to be mentioned,
  • are self-healing effects in those materials possible?
  • "6. Summary" contains some recommendations, so it could be re-named as "Summary and Future Outlooks"; for future developments one could briefly show the role of various inorganic fillers that improve the properties of epoxy coatings.
  • Please check Ref. [28] - the authors are   C. Carfagna and  A. Apicella.

Author Response

Comment 1: in "Abstract: "... can dynamically develop new transport mechanisms" - what you mean by "new mechanisms" - are they not known in chemical engineering? "This review examines common analytical and spectroscopic techniques..." - are spectroscopic techniques not analytical?

Author response: Thank you for pointing this out. The reviewer is correct, and we have changed “new mechanisms” to “different transport mechanisms”. Also, “spectroscopic” has now been removed.

Comment 2: it is a vast work - Table of Contents on the beginning would help the reader to get familiar with the scope; moreover, list of abbreviations is recommended,

Author response: A Table of Contents and a list of abbreviations are now added (after Abstract and before References, respectively).

Comment 3: anticorrosive functions in the epoxy coatings deserve to be mentioned,

Author response: Anticorrosive performance has been addressed at line 9 of the second paragraph of the Introduction in the revised manuscript. The added section to the manuscript is as follows:

“For instance, the peripheral polar functional groups in the epoxy act as adsorption sites for inhibitors (e.g., amines, thiols, and alcohol-based epoxy resins), which enhances its anticorrosive effects in aqueous media when applied as a coating [14]. In addition to the main macromolecular compound, organic and inorganic additive components can be used to improve the anticorrosive performance of the coating [15].”

Comment 4: are self-healing effects in those materials possible?

Author response: The possibility of incorporating self-healing components in powder coatings has been addressed at line 18 of the second paragraph of the Introduction in the revised manuscript. The added content to the manuscript is as follows:

“Microencapsulated agents―also known as self-healing components―may also be used as additives.  These provide a polymer mending property in case of mechanical damage to the coating and may effectively protect the coated steel surface from corrosion [17], [18].”

Comment 5: "6. Summary" contains some recommendations, so it could be re-named as "Summary and Future Outlooks"; for future developments one could briefly show the role of various inorganic fillers that improve the properties of epoxy coatings.

Author response: The title of the last section has been changed accordingly.

Comment 6: Please check Ref. [28] - the authors are   C. Carfagna and  A. Apicella.

Author response: This correction has been made in the revised bibliography.

Reviewer 2 Report

Review (recommended major revision)

Subject is appropriate for the scope of journal. Article is well written with clear research points, the explanations of relevant reviewed ideas and discussion. Idea is well-thought out, well-defined and relevant. Information is collected in a structured way.

***uploaded review file***

Author Response

Comment 1: “Development of an organic coating requires compatibility with specific environmental regulations and safety concerns [7]; the coating molecular structure has been modified in some cases to meet new expectations [8] [10]. A necessity to increase the bio-based material content in coatings is completely lacking in the review at hand, but should be addresses. Some recent works include 10.1515/revce-2019-0077, 10.1038/s41598-020-67787-9, and 10.1021/acssuschemeng.0c06099, but at least a very short chapter should be dedicated to this, which would be quite welcome.

Author response: The necessity of considering bio-based material content in coatings has been added to the second paragraph of the Introduction (2nd line) in the revised manuscript and all suggested references are cited accordingly. The added section to the manuscript is as follows:

“There are many proposed new coating systems for petroleum-based products and, since climate change is a serious concern, recent trends in alternative fuels suggest that there is a necessity to increase the bio-based material content in coatings [14]–[16]. However, the pipeline industry continues to use epoxy-based coatings. Some functional fillers such as needle-shaped amorphous wollastonite are used to increase flexural modulus and lower the thermal expansion and shrinkage of the final coating [17], [18]. “

Comment 2: The conclusions are interesting and the in-depth research meticulous. Results provide coherent summary and suggestion for future work and analyses.

Author response: Thank you! The title of the last section has been changed to "Summary and Future Outlooks".

Comment 3: Grammar is appropriate and understandable, although some minor revision may be needed concerning the tenses used in text (Present and Past used interchangeably in Introduction).

Author response:

The introduction has been updated to use the present tense throughout.

Comment 4: Some figures have a different alignment and size should be uniform throughout the article (fig. 2, 3, 7, table 7)

Author response: These formatting issues are related to the transformation of the manuscript by the journal. We kindly ask the editor to consider this issue for the revised article.

Comment 5: Table 5 revise the different font size in the caption

Author response: These formatting issues are related to the transformation of the manuscript by the journal. We kindly ask the editor to consider this issue for the revised article.

Comment 6: Two articles in references not referenced in text [16], [17].

Author response: This is due to a referencing software malfunction; the issue has been resolved in the revised manuscript.

Reviewer 3 Report

The paper fits well with the journal requirements. By my side, it could be considered for a publication.

Best wishes of happy Easter.

Author Response

Comment: The paper fits well with the journal requirements. By my side, it could be considered for a publication.

Author response: Thank you!

Round 2

Reviewer 2 Report

Review (accept)

Comments have been mostly accepted, improving this interesting review, which will prove as relevant to the community at large.